# Measuring the impact of the Capital Card®, a novel form of contingency management, on substance misuse treatment outcomes: A retrospective evaluation

**Antony C. Moss**[1]*, **Devon De Silva**[2], **Sharon Cox**[1], **Caitlin Notley**[1,3], **Manish Nanda**[2]

**1** Centre for Addictive Behaviours Research, School of Applied Sciences, London South Bank University, London, United Kingdom, **2** Innovation Research Unit, WDP, London, United Kingdom, **3** Norwich Medical School, University of East Anglia, Norwich, United Kingdom

* mossac@lsbu.ac.uk

**Data Availability Statement:** Tthe underlying data came from two sources – the National Drug Treatment Monitoring Service, and WDP's own

## Abstract

### Background

The Capital Card, developed by WDP, is a digital innovation which acts as a form of contingency management, and aims to significantly improve service user outcomes. WDP is a substance misuse treatment provider commissioned by local authorities across the UK to support service users and their families affected by addiction. The Capital Card, much like commercial loyalty cards, uses a simple earn-spend points system which incentivises and rewards service users for engaging with services e.g. by attending key work sessions, Blood Borne Virus appointments or group-work sessions. The Spend activities available to service users are designed to improve overall wellbeing and build social and recovery capital, and include activities such as educational classes, fitness classes, driving lessons, and cinema tickets.

### Methods and findings

We compared successful completion rates of 1,545 service users accessing one of WDP's London based community services over a two-year period; before and after the Capital Card was introduced. Client demographics (age, sex and primary substance) were controlled for during the analysis. Once client demographics were controlled for, analysis showed that clients with a Capital Card were 1.5 times more likely to successfully complete treatment than those who had not had the Capital Card (OR = 1.507, 95% CI = 1.194 to 1.902).

### Conclusions

The results of this initial evaluation are of particular interest to commissioners and policy makers as it indicates that the Capital Card can be used effectively as a form of contingency management to enhance recovery outcomes for service users engaging in community-based substance misuse services.

Capital Card Scheme records. We do not have permission to share the NDTMS portion of the dataset online, as this is controlled by Public Health England, but the NDTMS do allow individuals to request access to data for research purposes (contact details for this are provided in our original submission). With regards to the data controlled by WDP, if an individual were to obtain the data from the NDTMS for the purposes of replicating our analyses, a request for the anonymised Capital Card data could be made to the Innovation and Research Unit (IRU@wdp.org.uk) who would be able to match client records, and provide a file in an anonymised format.

**Funding:** The author(s) received no specific funding for this work.

**Competing interests:** WDP owns the intellectual property for the Capital Card. Manish Nanda and Devon De Silva are employees of WDP. This does not alter our adherence to PLOS ONE policies on sharing data and materials.

## Introduction

Contingency management (CM) interventions for substance misuse have been developed and tested across a range of substances, starting at least as early as the 1960s (e.g. [1]). CM interventions are based on basic learning principles of operant conditioning, whereby rewards (in the form of positive or negative reinforcement) and punishments are used to promote behaviours which are likely to improve treatment outcomes, and discourage behaviours which may lead to poorer treatment outcomes. CM interventions are widely regarded as effective in helping to improve recovery outcomes across a range of substances (e.g. [2–4]). CM is especially valuable as an adjunct treatment, with some evidence suggesting this is associated with an improvement in treatment outcomes for those individuals who may not be strongly motivated to address their substance misuse for a variety of reasons (e.g. [5]).

The adoption of CM, despite a strong evidence base, has been inconsistent across treatment services. For example, one study in 2004 found that fewer than 10% of front-line staff reported using CM as part of their routine structured treatment for substance misuse [6]. In a trial exploring the implementation of CM amongst front line staff trained in its use, it was found that only 58% reported doing so at any point during a three month follow up period, with this percentage falling to 25% amongst therapists who saw fewer substance -abusing clients [7]. The barriers to adoption of CM are complex and, in McGovern et al.'s [6] study, range from practitioner-level issues (e.g. moral objections, or lack of expertise in CM), to organisational barriers (e.g. lack of funding for CM, lack of demand from funders for CM to form part of commissioned services), and implementation can even be influenced by perceived characteristics of the adult seeking treatment as being more or less 'appropriate' for CM-type approaches [8]. In a wider context, the low rates of uptake of CM for substance misuse may be a consequence of what has been termed a 'moral sidestep' in drug policy [9], whereby a range of effective interventions may fail to receive political support due to their incompatibility with (often) conservative political values.

Nevertheless, CM is associated with positive outcomes and has been proven cost-effective. Indeed, in a cost effectiveness analysis, it was shown that CM using a larger reward was in fact more cost effective than a smaller reward [10]. In the UK, funding for substance misuse treatment services has been reduced over recent years [11], although the number of people estimated in need of drug and alcohol support has not [12]. Substance misuse services need to investigate all avenues of potential benefit for their service users and CM is one of those.

The Capital Card Scheme is a novel form of contingency management, perhaps more closely aligned with Voucher Based Reinforcement Therapy [13], in that service users accrue points for engaging with their structured treatment, and these points can then be spent on a range of activities. Unlike most forms of contingency management, the Capital Card Scheme has been developed to operate across the entirety of a substance use treatment service, and so is not restricted to specific populations, substance users, or treatment types. The scheme itself relies solely on positive reinforcement (the awarding of points for any form of engagement with the service), and does not utilise negative reinforcement, or positive punishment, which has been used in other implementations of contingency management. As an innovative approach to utilising digital technologies to improve treatment outcomes, the Capital Card Scheme won the *Digital Innovation of the Year* award at the UK's *Third Sector Excellence Awards 2018*.

Here we present the results of a retrospective analysis of routinely collected treatment service outcome data reported over a two-year period, covering the years before and after the Capital Card Scheme was introduced across an entire substance use treatment service in one London borough. The purpose of this analysis was to determine whether the introduction of

the Capital Card was associated with changes in successful treatment completion, after controlling for differences between the service users in these two time periods.

## Method

Our study involved the secondary analysis of data held in a government and so no specific ethical approval was obtained for the study. The data analysed were shared with the authors in line with existing consent protocols for the database being accessed, within which it is understood by participants that their data may be anonymously analysed for audit and research purposes.

### Participants

All service users engaging with structured treatment in the London Borough of Hackney between 1st April 2016 and 31st March 2018 were included in this retrospective data analysis. This provided for a two-year period, wherein the Capital Card was introduced half way through for all service users, and initially included 2,722 service user case records. Inclusion criteria for this analysis were that services users had to be 18 years of age or older, and have consented to a treatment outcome recorded in the National Drug Treatment Monitoring Service (NDTMS) database. All substance misuse treatment providers within the UK are required, where consent is given, to submit certain anonymised service user information to Public Health England (PHE, an executive agency of the UK's Department of Health and Social Care who are responsible for substance misuse services) [12]. This information allows PHE, commissioners and local treatment providers to monitor national and local trends and respond accordingly. Data used in this analysis were provided in a fully anonymised format by WDP.

Service users recorded in the NDTMS who were still undergoing treatment after 31st March 2018 were excluded (n = 908), as we only included those with a recorded treatment outcome in the analysis. In addition, service user records were excluded where the NDTMS record showed a Transfer (n = 235), as we were unable to reliably determine treatment outcomes for this group. Of the 235 who had a Transfer status, 120 were being held in custody, and 115 had been transferred to another treatment service. A further 25 service users who passed away while in treatment were excluded from the analysis. Finally, nine service users were excluded who had declined treatment commencement, and so had not engaged with any structured intervention.

Inclusion in the analysis required service users to have a recorded treatment outcome within the time period of the study. The definition of a successful treatment outcome for the purpose of the analysis presented here included (per NDTMS coding conventions): Treatment complete–alcohol free; Treatment completed–drug free; and Treatment completed–occasional user (not opiates or crack). Unsuccessful treatment outcomes included: Incomplete–dropped out; and Incomplete–treatment withdrawn by provider.

Table 1 provides a breakdown of all 1,545 service users who were identified for inclusion through this time period and breaks down the demographic and substance use characteristics across the pre- and post-intervention periods. Analysis of demographic data shows there was no significant difference in primary substance or housing status between these two intervention periods, but significant differences in age and profile of gender were observed–service users in the Capital Card period were older, and there were proportionally more female services users in the Capital Card period. Table 2 illustrates the percentage of service users who recorded a successful treatment outcome across both groups, and overall.

### Capital Card scheme

All service users engaging in structured treatment in a WDP service where the Capital Card is live are provided a Capital Card following completion of a full, comprehensive

**Table 1. Breakdown of demographic characteristics of service users included in the analysis.**

| | No Capital Card (%) | Capital Card (%) | Total (%) |
|---|---|---|---|
| *Gender* | | | |
| Female | 312 (28) | 160 (37) | 472 (31) |
| Male | 804 (72) | 269 (63) | 1073 (69) |
| *Primary Substance* | | | |
| Alcohol | 472 (42) | 202 (47) | 674 (44) |
| Opiate | 292 (26) | 120 (28) | 412 (27) |
| Non-Opiate | 352 (32) | 107 (25) | 459 (29) |
| *Housing Status* | | | |
| No Housing Problems | 762 (76) | 292 (72) | 1054 (75) |
| Housing Problems | 171 (17) | 80 (20) | 251 (18) |
| No Fixed Abode | 73 (7) | 33 (8) | 106 (7) |

assessment. Service users were also offered a companion app which allowed them to monitor their earn/spend activity and provides them with an electronic version of their Capital Card. The physical or electronic Capital Card was then also used by service users to 'tap-in', an electronic log of their attendance which provided points each time they engaged in a recovery related activity. Examples of such activities for which points could be earned are attending key work sessions, attending bloodborne virus testing appointments, engaging in group-work sessions, participating in research projects being conducted by the treatment provider, and engaging with harm reduction interventions such as naloxone training or attending needle exchanges. Once enough points have been accrued, these can be spent at local community services and businesses which are working in collaboration with the service provider on this initiative. The spending activities available to service users were also chosen as ones which may also assist in improving overall wellbeing (e.g., fitness and cooking classes, education services) and build social and recovery capital through engagement with both the recovery process and also the local community. While there is no direct cash equivalent for points earned in the scheme, examples of the rewards and the points required include two cinema tickets for 80 points, an annual gym membership for 100 points, a free bicycle service for 100 points, and £5 off of driving lessons for all Capital Card holders (i.e. no points are required to redeem this discount).

The arrangement with local retailers and businesses is that they are effectively making charitable donations to the Capital Card Scheme to support service users. WDP staff time to support the scheme is supported as part of contracts with individual local authorities. As noted above, while some partner organisations are unable to offer goods and services for free, they are often willing to offer discounts to service users.

## Service user involvement

A focus group, attended by Hackney service users, was held during the initial design phase and feedback, such as ensuring that all earn activities are worth the same amount of points, was

**Table 2. Successful treatment completions (%) by primary substance and Capital Card group.**

| Primary Substance | No Capital Card | Capital Card | Total |
|---|---|---|---|
| Alcohol | 51 | 71 | 57 |
| Opiate | 47 | 54 | 49 |
| Non-Opiate | 26 | 34 | 28 |

taken on board as part of the developed of the Capital Card scheme. WDP continue to ascertain service user satisfaction have received positive feedback from service users, such as:

*Back up north where I'm from there's no set up like the Capital Card. I don't think people in London realise how lucky they are to have this scheme, there are so many activities to do. It's the first time in years I've felt optimistic about my future. It's just brilliant."*

*(Male, 45yrs)*

*"I am happy to be able to socialise with my peers rather than isolate at home"*

*(Female, 52yrs)*

Service users also continue to be involved in the ongoing development of the scheme by providing the Capital Card team regular feedback on what is working well and what could be improved. Below are examples of such feedback, which has been acted upon by the Capital Card team:

*"Request for Christmas pop-up shop"*

*(Anonymous feedback)*

*"Not enough men's clothing at the Capital Card shop"*

*(Male, 64yrs)*

In addition, service users are also able to become either Capital Card Ambassadors, which involves promoting the scheme externally, or Capital Card Champions, which involves promoting the scheme internally and sharing their experiences with other service users. This involves the provision of training for those service users who wish to become involved as either ambassadors or champions, providing an important service user-led focus in the ongoing roll out and development of the scheme, while also providing ambassadors and champions with important transferable employability skills. Positive feedback for this aspect of the scheme has been received, with one ambassador commenting:

*"I'm now doing Capital Card Ambassador training, where I will be able to go out and talk to local businesses about the Capital Card, helping to grow the scheme. It's small at the moment, but I really believe in the idea and have seen the motivation and nudge it gives people here. "*

*Peer Support Mentor and Capital Card Ambassador.*

## Results

To determine whether there were any significant differences in the profile of service users across the two time periods, we conducted a series of inferential analyses comparing age, gender, primary substance of use, and housing status. We attempted to look at differences in self-reported mental health issues, but due to changes in NDTMS coding during the time period, were unable to reliably extract these data to compare across the groups.

An independent samples t-test demonstrated that the No Capital Card group were significantly younger (M = 39.8 yrs, SD = 11.1) than the Capital Card group (M = 41.6, SD = 9.9), t (1,543) = 2.95, p < .01. A chi-square analysis demonstrated a significant association between

group and gender, $\chi^2$ (1) = 12.7, p < .001, with a higher proportion of females in the Capital Card period. Similarly, a significant association between group and primary substance (categorised as alcohol, opiate, and non-opiates) was identified, $\chi^2$ (2) = 6.55, p < .05. An inspection of the adjusted residuals revealed that this association was due to a higher proportion of non-opiate users in the No Capital Card period (p < .05), with no significant differences in the proportion of alcohol or opiate users across the two groups. Finally, no association was found between housing status (using the categories No Housing Problem, Housing Problem, No Fixed Abode), $\chi^2$ (2) = 2.05, p > .05.

## Capital Card points earned and spent

On average, service users earned 92 Capital Card Points (SD = 127, range = 0–890), and spent an average of 15 points (SD = 43, range = 0–320). Only 21 of the 429 service users in the Capital Card period, accrued zero points. This means that 95.1% of service users engaged with the scheme and accrued some points during their treatment. However, 368 (85.8%) service spent none of their points earned during this period. Service users who spent no points accrued significantly fewer points (M = 59, SD = 77) than those who spent points (M = 282, SD = 164), t (427) = 16.5, p < .001.

## Impact of the Capital Card Scheme on treatment outcomes

To test whether the implementation of the Capital Card was associated with a change in treatment outcomes across these two time periods, a hierarchical binary logistic regression was conducted, with treatment outcome (successful completion vs. unsuccessful) as the criterion variable, and Capital Card group as a categorical predictor. To control for the group differences identified above, age was entered as a continuous predictor, along with gender and primary substance (using alcohol as the reference category, as the largest subgroup) as categorical predictors in Step 1 of the model. Capital Card group was then entered at Step 2. The overall model was significant at Step 1, $\chi^2$ (4) = 89.5, p < .001. Table 3 shows that neither age nor gender predicted treatment outcome, but that opiate and non-opiate users were less likely to be in the successful treatment completion group than alcohol users. After Capital Card group was entered at Step 2, the overall model remained significant, $\chi^2$ (5) = 101.5, p < .001. Examination of the co-efficients in Table 2 reveal that primary substance remained a significant predictor of outcome, and that access to the Capital Card was associated with a 1.5 times greater likelihood of positive treatment outcomes (OR = 1.507, 95% CI = 1.194 to 1.902). An interaction term with Capital Card x Primary Substance was included to determine whether there was a selective effect of the intervention by primary substance, but this was not significant.

## Impact of the Capital Card Scheme on quality of life and psychological wellbeing

We compared change in both Quality of Life and Psychological Wellbeing measures across the two time periods, to determine whether availability of the Capital Card was associated with improvements in either of these two measures. The sample included in this analysis was, by necessity, restricted only to those service users who completed their treatment, and where both psychological wellbeing and quality of life measures had been obtained both at entry to structured treatment, and on exit from treatment. This provided a sample of 567 services users (371 in the No Capital Card group, and 196 in the Capital Card group). Independent samples t-tests showed no significant difference in change scores for either psychological wellbeing or quality of life–with both measures improving between treatment start and treatment exit for all groups (see Table 4).

**Table 3. Hierarchical binary logistic regression predicting probability of successful treatment completion.**

|  | B | S.E. | Wald | df | Sig. | Exp(B) | Exp(B) 95% CI | |
|---|---|---|---|---|---|---|---|---|
|  |  |  |  |  |  |  | Lower | Upper |
| Step 1 |  |  |  |  |  |  |  |  |
| Age | -.006 | .005 | 1.56 | 1 | .212 | .994 | .984 | 1.004 |
| Gender | -.006 | .115 | .002 | 1 | .961 | 1.006 | .803 | 1.260 |
| *Primary Substance* |  |  |  |  |  |  |  |  |
| Alcohol (Reference category) |  |  | 82.46 | 2 | .000*** |  |  |  |
| Opiate | -1.23 | .137 | 82.00 | 1 | .000*** | .290 | .222 | .379 |
| Non-Opiate | -.37 | .127 | 8.28 | 1 | .004** | .694 | .541 | .890 |
| Step 2 |  |  |  |  |  |  |  |  |
| Age | -.007 | .005 | 2.18 | 1 | .140 | .993 | .983 | 1.002 |
| Gender | -.029 | .116 | .063 | 1 | .801 | .971 | .774 | 1.219 |
| *Primary Substance* |  |  |  |  |  |  |  |  |
| Alcohol (Reference category) |  |  | 83.56 | 2 | .000*** |  |  |  |
| Opiate | -1.25 | .137 | 82.90 | 1 | .000*** | .287 | .219 | .375 |
| Non-Opiate | -.35 | .128 | 7.64 | 1 | .006** | .703 | .548 | .903 |
| Capital Card | .41 | .119 | 11.92 | 1 | .001*** | 1.507 | 1.194 | 1.902 |
| Capital Card*Primary Substance |  |  | 4.40 | 2 | .111 |  |  |  |

*p < .05;

**p < .001;

***p < .001.

## Discussion

The aim of this analysis was to determine whether the introduction of a novel contingency management intervention, the Capital Card, was associated with a change in treatment outcome rates. After controlling for differences in demographic characteristics across the two time periods involved in this analysis, the Capital Card was associated with an improved likelihood of services users successfully completing treatment. The analysis also demonstrated that there was no significant difference in terms of changes to Quality of Life or Psychological Wellbeing, which generally improved for all service users who successfully completed their structured treatment. This data analysis provides preliminary evidence supporting the use of the Capital Card, a service-wide approach to contingency management, as a means of improving treatment outcomes for those seeking treatment in substance use services within this local context.

The primary limitation of this analysis is that it is based on a retrospective analysis of routinely collected treatment outcome data, and is therefore not a definitive trial. This meant our analyses were constrained to only those variables which were recorded at the time, and variables such as addiction severity of socioeconomic status which may affect treatment outcomes were not available. We were also unable to control for pre-existing and comorbid mental

**Table 4. Quality of life and psychological health change scores.**

|  | No Capital Card | | Capital Card | |  |  |
|---|---|---|---|---|---|---|
|  | N | Mean (SD) | N | Mean (SD) | t (df = 565) | p |
| Quality of Life | 371 | 2.96 (11.16) | 196 | 2.96 (14.72) | .002 | .480 |
| Psychological Health | 371 | 3.39 (11.12) | 196 | 2.88 (15.92) | .445 | .178 |

health problems across the two time periods due to an absence of reliable data, and so cannot exclude the possibility that differences in treatment outcome might be attributable in whole or part to differences in the prevalence of mental health issues pre and post the introduction of the Capital Card. Finally, the present analysis does not provide any insight in to the process by which the Capital Card scheme itself may be improving treatment outcomes, beyond the assumptions we might make regarding contingency management interventions in general.

Notwithstanding these limitations, the present results do provide prima facie support for the Capital Card Scheme, and suggest the need for a more definitive evaluation of the intervention to establish its efficacy. Further, given the complexity of this intervention which is made available to all service users engaging in structured treatment, and which provides a wide range of rewards and ways of spending points earned, there are a number of questions to be explored from a process evaluation perspective. For example, it would be useful to understand how this intervention is being utilised by service users in different ways, to potentially meet a range of specific needs.

A further feature of the Capital Card Scheme which was not explored here is that the points earned during treatment can be spent for up to 12 months after exiting the service. Service users can also continue earning points for 6 months. In light of the relatively high proportion of service users who earned but did not spend any points, this raises an interesting question for further research as to whether the impact of the scheme is attributable to the activities which these points can be spent on, or whether there is any inherently motivating quality in earning the points themselves. As such, it would be useful to determine whether this impacts on longer term recovery outcomes, beyond the mere completion of a structured programme of treatment.

One final, important reflection on the analysis presented herein relates to insights which might be derived from the way in which service users appeared to engage with the Capital Card Scheme itself. While the scheme has been designed explicitly as a contingency management intervention, during the time period within which this retrospective analysis has been conducted, the majority of service users did not spend the majority of their points. As such, it might be seen that any positive impact of the scheme might be attributed to a form of token economy–that is to say, the mere collection of points through the scheme may have positively impacted treatment outcomes, beyond the impact of any actual rewards which may have been purchased with the points themselves. This is an important area for a future prospective evaluation of the scheme to explore–to understand the role that the accrual of points and their expenditure might play in helping support treatment completion, and whether collecting and spending may be more important for service users at different stages of their recovery.

## Author Contributions

**Conceptualization:** Antony C. Moss, Devon De Silva, Sharon Cox, Caitlin Notley, Manish Nanda.

**Data curation:** Antony C. Moss, Devon De Silva.

**Formal analysis:** Antony C. Moss, Sharon Cox.

**Methodology:** Antony C. Moss, Devon De Silva, Sharon Cox, Caitlin Notley, Manish Nanda.

**Writing – original draft:** Antony C. Moss.

**Writing – review & editing:** Antony C. Moss, Devon De Silva, Sharon Cox.

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
