## [Decision Letter · Decision Letter 0]

29 Aug 2019

PONE-D-19-18131

Measuring the impact of the Capital Card®, a novel form of contingency management, on substance misuse treatment outcomes: A retrospective evaluation

PLOS ONE

Dear Dr. Moss

Thank you for submitting your manuscript to PLOS ONE. After careful consideration, we feel that it has merit but does not fully meet PLOS ONE’s publication criteria as it currently stands. Therefore, we invite you to submit a revised version of the manuscript that comprehensively addresses all of the points raised during the review process. In particular, both reviewers raised concerned about the completeness of descriptions of samples, methodology and analyses, with Reviewer 2 making specific requests for additional analyses. Given the limitations in the design for supporting inferences of program effectiveness, and some question as to what degree the program is consistent with  CM, moreover,  greater emphasis should be put on the the study's preliminary nature in the Title, Abstract, Introduction, and in the Discussion. More careful wording of the results and their meaning  (e.g., null findings for group differences do not "demonstrate" no difference) and greater depth in the critical appraisal of the actual benefits of the Capital Card program are warranted. Finally, though the text and tables refer to gender, in fact I believe that this variable better represents sex, as gender roles and identity appear not to have been the basis for the dichotomous designation used.

We would appreciate receiving your revised manuscript by Oct 13 2019 11:59PM. To enhance the reproducibility of your results, we recommend that if applicable you deposit your laboratory protocols in protocols.io, where a protocol can be assigned its own identifier (DOI) such that it can be cited independently in the future. For instructions see: http://journals.plos.org/plosone/s/submission-guidelines#loc-laboratory-protocols

We look forward to receiving your revised manuscript.

Kind regards,

Thomas G. Brown, Ph.D.

Academic Editor

PLOS ONE

Journal Requirements:

2. In the ethics statement in the manuscript and in the online submission form, please provide additional information about the patient records used in your retrospective study. Specifically, please ensure that you have discussed whether all data were fully anonymized before you accessed them and/or whether the IRB or ethics committee waived the requirement for informed consent. If patients provided informed written consent to have data from their medical records used in research, please include this information.

"WDP owns the intellectual property for the Capital Card. Manish Nanda and Devon De Silva are employees of WDP."

Reviewers' comments:

Reviewer's Responses to Questions

**Comments to the Author**

1. Is the manuscript technically sound, and do the data support the conclusions?

Reviewer #1: Yes

Reviewer #2: Partly

2. Has the statistical analysis been performed appropriately and rigorously? 

Reviewer #1: Yes

Reviewer #2: Yes

3. Have the authors made all data underlying the findings in their manuscript fully available?

Reviewer #1: Yes

Reviewer #2: No

4. Is the manuscript presented in an intelligible fashion and written in standard English?

Reviewer #1: Yes

Reviewer #2: Yes

5. Review Comments to the Author

Reviewer #1: This is an unmasked review of a manuscript principally describing retrospective analysis of a Capital Card, likened by the authors to a commercial loyalty card, and its association with substance use treatment outcomes over a two-year period among 1545 persons from a United Kingdom borough. A limited set of demographic data were examined and controlled for in later analyses examining differences in treatment outcomes between those entering treatment programs in the year prior vs. following introduction of the Capital Card. Reported findings note that, after controlling for age and substance of abuse, those entering treatment programs following introduction of the Capital Card were more likely than their counterparts to complete treatment. While a primary study caveat (lack of randomization to comparison groups) is acknowledged by the submitting authors, it carries significant limitations and prevents meaningful contribution to extant addiction treatment literature. In particular, this quasi-experimental design precludes any control for likely influences of history as well as a host of other 3rd-variables (socioeconomic status, income, addiction severity, and treatment facility to name a few) oddly omitted from consideration despite author indication that all persons included in the analyses had completed a comprehensive intake assessment. Additional study caveats include: 1) inadequate description of the service data from which the categorical determination of treatment outcome occurred; 2) errant description of the intention of the Capital Card as a form of voucher-based contingency management, when in fact its description as a point system instead reflects something much more like a type of token economy; 3) absence of behavioral reinforcement for a large majority of Capital Card recipients, as evidenced by report that 85+% of Capital Card recipients never exchanged their ‘points’ for any tangible reinforcer, that calls into question whether this Capital Card functioned as a form of contingency management; and 4) the included suggestion of an ethically questionable provision that during the study period Capital Card recipients were concurrently recruited to promote use of the Capital Card ‘scheme’ to others.

Reviewer #2: This report describes a retrospective assessment of the effectiveness of an incentive program used in outpatients with substance use disorders. Outcomes obtained with 429 patients treated during the year after the incentive program was implemented were compared to those from 1,116 treated during the year before implementation (overall N= 1,545) in a single borough in London. After controlling for sample differences in socio-demographic characteristics and primary substance use disorder, odds of completing treatment were 1.5 times greater in the incentives compared to no-incentives conditions. Acknowledging the limitation of the research design for supporting causal inferences, the authors reasonably conclude that these results are sufficiently positive to warrant conducting a randomized controlled trial to prospectively assess the effectiveness of this incentive intervention.

As the authors appropriately note in the Introduction, dissemination of Contingency Management (CM) interventions into community substance abuse treatment services has been disappointing considering the enormous amount of empirical evidence from controlled studies supporting its efficacy. For that reason, the present report detailing what appears to be a successful implementation of CM into community treatment programs has the potential to be of interest to researchers, policy makers, and clinicians involved with prevention and treatment of substance use disorders. The report is generally well written, and the study appears to have been reasonably well conducted, and the results analyzed and interpreted appropriately. Enthusiasm is dampened to some extent by a lack of information in several areas detailed below. None of these matters are necessarily fatal flaws, but they should be addressed.

(1) The authors assume a familiarity with the service provider, WDP. I took the initiative to investigate online but readers should not need to do so. Please revise the opening sentence of the Abstract keeping in mind that many readers of this international journal will have no familiarity with WDP, nor “sector, first awards”, and may not understand what you mean by the parenthetical mention of the NICE clinical guideline on CM. Please revise or deleate that sentence. Please also add a couple of sentences to the Methods that provides readers with additional information on WDP and the services it provides. I can see why the authors may want to mention the sector first award, but that should only be mentioned once, either in the Intro or Discussion and with sufficient background for readers to understand its significance.

(2) Please add additional detail to the Methods on the following three aspects of the CM intervention. Say more using examples of they type of activities that earned points. Note the monetary value of points, monetary value of total possible earnings, whether number of points that could be earned varied by activity, and if possible the average monetary value of incentives earned in the present study. Lastly, please provide more information regarding the arrangement that WDP has with local retailers to provide incentives. Is the incentive program entirely funded through charitable donations from these retailers? Are NICE funds involved and if not, why not? Lastly, please move to the Methods the information that is now in the Discussion regarding patients being able to spend points for 12 months following treatment completion and to continue earning them for an additional 6 months. Those are important features of the program.

(3) Please provide additional breakdown of outcomes by primary type of SUD. You may consider adding an interaction term to the model on type of SUD. Even a descriptive breakdown on treatment completion rates before and following implementation of the incentive program by primary SUD would be helpful. Readers are surely going to be curious about potential differences.

(4) Please add parenthetical percentages to the tables so readers don’t have to do the calculations on their own as they read through the report, which is what I found myself doing.

6. PLOS authors have the option to publish the peer review history of their article (what does this mean?). If published, this will include your full peer review and any attached files.

Reviewer #1: No

Reviewer #2: No

---

## [Author Response · Author response to Decision Letter 0]

26 Nov 2019

Please see our Cover Letter and Response to Reviewer document, where we have provided detailed responses to all comments.

---

## [Editor Report · Decision Letter 1]

28 Nov 2019

PONE-D-19-18131R1

Measuring the impact of the Capital Card®, a novel form of contingency management, on substance misuse treatment outcomes: A retrospective evaluation

PLOS ONE

Dear Professor Moss,

Thank you for submitting your revised manuscript to PLOS ONE. After careful consideration, we feel that the revision does not fully meet PLOS ONE’s publication criteria as it currently stands. Therefore, we invite you to submit another  revision  of the manuscript that addresses the points raised during the second review process.

In particular, it is important for the authors to carefully consider Reviewer 2's comments, as his guarded recommendation to invite a revision was contingent on his recommendations being acted upon. Specifically, Reviewer 2's comment #1 requested more judicious use of unfamiliar acronyms. This has not  been completely addressed, as the meaning of PHE and WDP remain to be clarified. Please verify that all acronyms have been defined. More substantially, Reviewer  2's comment #3 has not been fully addressed adequately. Analyses summarized in Table 2 now regresses  primary drug, but this was not the intention of this comment. Inclusion as a main effect in regression speaks to drug profile general effects on outcome in step 1 with reference to alcohol's effect, and treatment effect re-estimated after accounting for this main effect in step 2. Its inclusion as an interaction term with treatment exposure (following its inclusion as a main effect as is now the case)  speaks to this CM variant's potential selective benefit for participants with different drug use profiles that was the point of of this comment. At the very least, as Reviewer 2 noted, a descriptive breakdown of outcome based upon drug subgroups and treatment exposure should be provided. Nevertheless, I believe that both approaches (descriptive and statistical) can and should be provided for better understanding the impact of treatment exposure on outcome in different drug using groups. Comment #4 has also been neglected, as the expression of statistics in Table 1 are limited to frequencies but not percentages as requested. 

We would appreciate receiving your revised manuscript by Jan 12 2020 11:59PM. To enhance the reproducibility of your results, we recommend that if applicable you deposit your laboratory protocols in protocols.io, where a protocol can be assigned its own identifier (DOI) such that it can be cited independently in the future. For instructions see: http://journals.plos.org/plosone/s/submission-guidelines#loc-laboratory-protocols

We look forward to receiving your revised manuscript.

Kind regards,

Thomas G. Brown, Ph.D.

Academic Editor

PLOS ONE

---

## [Author Response · Author response to Decision Letter 1]

10 Jan 2020

10th January 2020

Dear Dr Brown,

My co-authors and I are grateful for your feedback and for the opportunity to further revise and resubmit our manuscript. In response to the specific comments:

Specifically, Reviewer 2's comment #1 requested more judicious use of unfamiliar acronyms. This has not been completely addressed, as the meaning of PHE and WDP remain to be clarified. Please verify that all acronyms have been defined.

RESPONSE: we have further clarified the PHE acronym, as well as providing a brief explanation regarding the purpose of PHE as an organisation. With regards to WDP, this is in itself the name of the organisation, and not an acronym. WDP, as stated in the manuscript, is a substance misuse treatment provider.

More substantially, Reviewer 2's comment #3 has not been fully addressed adequately. Analyses summarized in Table 2 now regresses primary drug, but this was not the intention of this comment. Inclusion as a main effect in regression speaks to drug profile general effects on outcome in step 1 with reference to alcohol's effect, and treatment effect re-estimated after accounting for this main effect in step 2. Its inclusion as an interaction term with treatment exposure (following its inclusion as a main effect as is now the case) speaks to this CM variant's potential selective benefit for participants with different drug use profiles that was the point of this comment. At the very least, as Reviewer 2 noted, a descriptive breakdown of outcome based upon drug subgroups and treatment exposure should be provided. Nevertheless, I believe that both approaches (descriptive and statistical) can and should be provided for better understanding the impact of treatment exposure on outcome in different drug using groups.

RESPONSE: with regards to a descriptive summary, we did in fact include this as part of our previous revision (see Table 2). However, we accept the wider point that the inclusion of an interaction term in the regression model would provide additional clarity regarding potential differences in the effect of the intervention. We have now included this interaction term as requested, which in the event was not significant.

Comment #4 has also been neglected, as the expression of statistics in Table 1 are limited to frequencies but not percentages as requested.

RESPONSE: apologies if we have misunderstood here, but in the previous revision we did in fact add percentages in parentheses alongside the frequency data to this table. If further detail is required we are happy to provide this in the manuscript.

We are grateful for you taking time to consider our manuscript for inclusion in your special issue, and look forward to hearing from you in due course.

Yours sincerely,

Professor Antony C. Moss

Professor of Addictive Behaviour Science

Centre for Addictive Behaviours Research, London South Bank University, London, UK.

mossac@lsbu.ac.uk

---

## [Editor Report · Decision Letter 2]

7 Feb 2020

PONE-D-19-18131R2

Measuring the impact of the Capital Card®, a novel form of contingency management, on substance misuse treatment outcomes: A retrospective evaluation

PLOS ONE

Dear Professor Moss,

Thank you for submitting your manuscript to PLOS ONE. After careful consideration, we feel that it has merit but does not fully meet PLOS ONE’s publication criteria as it currently stands. Therefore, we invite you to submit a revised version of the manuscript that addresses the points raised during the review process.

We would appreciate receiving your revised manuscript by Mar 23 2020 11:59PM. To enhance the reproducibility of your results, we recommend that if applicable you deposit your laboratory protocols in protocols.io, where a protocol can be assigned its own identifier (DOI) such that it can be cited independently in the future. For instructions see: http://journals.plos.org/plosone/s/submission-guidelines#loc-laboratory-protocols

We look forward to receiving your revised manuscript.

Kind regards,

Dan Small, Ph.D.

Academic Editor

PLOS ONE

Additional Editor Comments (if provided):

This retrospective study of addiction treatment outcomes in relation to contingency management provides additional academic data to the expansive literature in the area. It’s originality, in my view, meriting publication is earned by two contributions. First, its focus on a particular digital innovation, the Capital Card, which appears to mirror “points cards” in use by people within everyday life, is relevant and novel. Secondly, the authors make several reflective comments about weaknesses in their research and, as such, thoughtfully point the way forward for future research. In particular, the authors astutely observe that their preliminary study was not able to measure some important variables such as the impact of socioeconomic variables such as homelessness and poverty. These are extremely significant points and merit careful academic consideration.

Moreover, they suggested that a future research study could be developed to more precisely evaluate contingency management (perhaps with a randomized control trial) so that some of the nuances could be uncovered. Some of the very thoughtful questions raised by the authors include:

• what accounts for why people do or don’t spend their points?

• what is the precise mechanism that accounts for the impact of contingency management?

• are the findings an artifact of a token economy?

Overall, it is my sense that the authors have effectively responded to the revisions requested by the various reviewers. As such, I recommend that the paper be published with three minor revisions.

1. There appears to be a minor typographical error in line 58. I believe that “…in tis use, it was…” should read: “…in its use, it was…”

2. There appears to be a word missing in line 228. It currently reads: “However, 368 (85.8%) service spent…” I believe that it should read: “However, 368 (85.8%) service users spent…”

3. There appears to be a typographical error on line 313 which currently reads: “did not spend he majority.” I believe that it should read: “…did not spend the majority.”

My apologies if the authors have already caught these three minor errors (they likely have); I’ve read all the versions of the manuscripts and perhaps I’ve missed the correction.

I believe that this study is a ground clearing exercise for a more in-depth research program by the authors. As such, I would add three questions for their future consideration as they contemplate a future research plan. Firstly, why does contingency management not improve quality of life or psychological well-being? Secondly, why are there differences in the efficacy of this model with respect to primary substance of choice by drug users? Finally, and perhaps most importantly, I want to encourage the authors to build on their question about the impact of this intervention on long-term recovery. It is interesting to consider that treatment, and the operant conditioning deployed here, reflects only a tiny part of the overall experience of the individual in recovery. Drug use, addiction and recovery do not really take place in clinics or recovery programs. They are part of a larger healing process that takes place within the wider, and much larger, lifeworld of which treatment is only a tiny part. How does this treatment tool, in the recovery tool belt, playout beyond the bounds of a healthcare environment?

---

## [Editor Report · Decision Letter 3]

19 Feb 2020

Measuring the impact of the Capital Card®, a novel form of contingency management, on substance misuse treatment outcomes: A retrospective evaluation

PONE-D-19-18131R3

Dear Dr. Moss,

We are pleased to inform you that your manuscript has been judged scientifically suitable for publication and will be formally accepted for publication once it complies with all outstanding technical requirements.

With kind regards,

Dan Small, Ph.D.

Academic Editor

PLOS ONE

Additional Editor Comments (optional):

Dear Professor Moss:

Thank for submitting the latest manuscript and for attending to the minor revisions as indicated. As such, I see the article as publishable.

Good luck with your future research in this area.

Kind regards,

Dr. Dan Small, PhD, MPhil
---

## [Editor Report · Acceptance letter]

25 Feb 2020

PONE-D-19-18131R3 

Measuring the impact of the Capital Card®, a novel form of contingency management, on substance misuse treatment outcomes: A retrospective evaluation 

Dear Dr. Moss:

I am pleased to inform you that your manuscript has been deemed suitable for publication in PLOS ONE. Congratulations! Your manuscript is now with our production department. 

With kind regards,

on behalf of

Dr. Dan Small 

Academic Editor

PLOS ONE